# Natriuretic Peptide Expression and Function in GH3 Somatolactotropes and Feline Somatotrope Pituitary Tumours

**DOI:** 10.3390/ijms22031076

**Published:** 2021-01-22

**Authors:** Samantha M. Mirczuk, Christopher J. Scudder, Jordan E. Read, Victoria J. Crossley, Jacob T. Regan, Karen M. Richardson, Bigboy Simbi, Craig A. McArdle, David B. Church, Joseph Fenn, Patrick J. Kenny, Holger A. Volk, Caroline P. Wheeler-Jones, Márta Korbonits, Stijn J. Niessen, Imelda M. McGonnell, Robert C. Fowkes

**Affiliations:** 1Endocrine Signalling Group, Royal Veterinary College, University of London, Royal College Street, London NW1 0TU, UK; samantha.byers@admin.cam.ac.uk (S.M.M.); cscudder@rvc.ac.uk (C.J.S.); j.read@qmul.ac.uk (J.E.R.); vcrossley@rvc.ac.uk (V.J.C.); jacobtcregan@icloud.com (J.T.R.); karen_richardson@me.com (K.M.R.); 2Comparative Biomedical Sciences, Royal Veterinary College, University of London, Royal College Street, London NW1 0TU, UK; bsimbi@rvc.ac.uk (B.S.); cwheeler@rvc.ac.uk (C.P.W.-J.); imcgonnell@rvc.ac.uk (I.M.M.); 3Clinical Sciences & Services, Royal Veterinary College, Hawkshead Lane, North Mymms, Hatfield, Hertfordshire AL9 7TA, UK; dchurch@rvc.ac.uk (D.B.C.); jfenn@rvc.ac.uk (J.F.); pkenny@sashvets.com (P.J.K.); holger.volk@tiho-hannover.de (H.A.V.); sniessen@rvc.ac.uk (S.J.N.); 4Department of Translational Science, Bristol Medical School, University of Bristol, Whitson Street, Bristol BS1 3NY, UK; craig.mcardle@bristol.ac.uk; 5Small Animal Specialist Hospital, 1 Richardson Place, North Ryde, 2113 NSW, Australia; 6Department of Small Animal Medicine and Surgery, University of Veterinary Medicine Hannover, Bünteweg 9, 30559 Hannover, Germany; 7Centre for Endocrinology, William Harvey Research Institute, Barts and the London School of Medicine and Dentistry, Queen Mary University of London, London EC1M 6BQ, UK; m.korbonits@qmul.ac.uk

**Keywords:** CNP, multiplex RT-qPCR, pituitary, somatotrope, acromegaly, feline

## Abstract

Patients harbouring mutations in genes encoding C-type natriuretic peptide (CNP; *NPPC*) or its receptor guanylyl cyclase B (GC-B, *NPR2*) suffer from severe growth phenotypes; loss-of-function mutations cause achondroplasia, whereas gain-of-function mutations cause skeletal overgrowth. Although most of the effects of CNP/GC-B on growth are mediated directly on bone, evidence suggests the natriuretic peptides may also affect anterior pituitary control of growth. Our previous studies described the expression of *NPPC* and *NPR2* in a range of human pituitary tumours, normal human pituitary, and normal fetal human pituitary. However, the natriuretic peptide system in somatotropes has not been extensively explored. Here, we examine the expression and function of the CNP/GC-B system in rat GH3 somatolactotrope cell line and pituitary tumours from a cohort of feline hypersomatotropism (HST; acromegaly) patients. Using multiplex RT-qPCR, all three natriuretic peptides and their receptors were detected in GH3 cells. The expression of *Nppc* was significantly enhanced following treatment with either 100 nM TRH or 10 µM forskolin, yet only *Npr1* expression was sensitive to forskolin stimulation; the effects of forskolin and TRH on *Nppc* expression were PKA- and MAPK-dependent, respectively. CNP stimulation of GH3 somatolactotropes significantly inhibited *Esr1, Insr* and *Lepr* expression, but dramatically enhanced *cFos* expression at the same time point. Oestrogen treatment significantly enhanced expression of *Nppa, Nppc, Npr1,* and *Npr2* in GH3 somatolactotropes, but inhibited CNP-stimulated cGMP accumulation. Finally, transcripts for all three natriuretic peptides and receptors were expressed in feline pituitary tumours from patients with HST. *NPPC* expression was negatively correlated with pituitary tumour volume and *SSTR5* expression, but positively correlated with *D2R* and *GHR* expression. Collectively, these data provide mechanisms that control expression and function of CNP in somatolactotrope cells, and identify putative transcriptional targets for CNP action in somatotropes.

## 1. Introduction

The natriuretic peptides (Arial, B-type, and C-type natriuretic peptides; ANP, BNP, CNP) have been characterised extensively in multiple species [1,2,3], where they have been shown to act as regulators of the cardiovascular system. Whilst ANP and BNP act predominantly to regulate cardiac function and natriuresis, it is clear that CNP exerts multiple effects throughout the body, acting as an autocrine/paracrine regulator in many endocrine tissues [2]. The majority of these effects are mediated via the receptor guanylyl cyclases (GC-A (for ANP and BNP), and GC-B (for CNP)), leading to the generation of cyclic guanosine 3′,5′ monophosphate (cGMP) and activation of protein kinase G (PKG) [3].

CNP is the major natriuretic peptide of the central nervous system [1,4], but has profound effects on growth in peripheral tissues. Targeted disruption of the *Nppc* or *Npr2* genes (encoding CNP and GC-B, respectively) lead to achondroplasia, dwarfism, female infertility and early death [5,6], and these severe growth phenotypes in mice are mirrored by the symptoms of human patients with inactivating mutations in *NPR2* and *NPPC* mutations [7,8]. Biochemical analyses of *Nppc*- and *Npr2*-disrupted mice indicate deficiencies in both growth hormone (GH) and insulin-like growth factor-1 (IGF-I) [5,6], which suggests a potential pituitary-specific role of CNP and GC-B signalling. Our previous work has shown that both CNP and GC-B are expressed in fetal and adult pituitary tissue in humans [9]. Furthermore, CNP appears to be the most potent natriuretic peptide in terms of cGMP generation in the pituitary, as demonstrated in a range of pituitary-derived cell lines [10,11]. Additionally, GC-B signalling is sensitive to heterologous desensitisation in gonadotroph-derived αT3-1 cells, and somatolactotrope-derived GH3 cells, following exposure to either gonadotropin-releasing hormone (GnRH) [10] or thyrotropin-releasing hormone (TRH) [12]. However, the local regulation and function of CNP in somatotrophs remains to be determined.

Pituitary tumours represent the most common form of intracranial tumour in humans [13]. Our previous studies have shown the expression of a natriuretic peptide system in a range of human pituitary tumours, regardless of the cell type of origin [9]. Recently, we have established that the prevalence of acromegaly (or hypersomatotropism, HST) in the domestic cat (*Felis catus*) are approximately ten times that seen in the human population, and is a primary cause of diabetes mellitus in 25% of diabetic cats [14]. Molecular and histological analyses of these feline pituitary tumours reveals expression of therapeutic targets, such as somatostatin and dopamine receptors [15]. However, the natriuretic peptide system in feline pituitaries has yet to be investigated.

Anterior pituitary regulation of growth is exquisitely controlled by various hormones, such as growth hormone-releasing hormone (GHRH), somatostatin (SST), TRH, and oestrogen [16,17,18,19,20]. Although the roles of these endocrine factors in regulating somatotroph behaviour are well established, there is increasing evidence that a local system of control exists within the pituitary [21], a phenomenon of so-called paracrinicity. Previous studies suggest that both CNP and cGMP are capable of regulating GH secretion [22,23], and long-acting CNP analogues are currently in therapeutic use to treat short stature patients, having been shown to improve growth in models of dwarfism and skeletal dysplasias [24,25,26,27]. In our current study, we use a customised multiplex RT-qPCR assay to examine the gene expression of multiple transcripts involved in the natriuretic peptide pathways, somatotrope markers, and receptors for hypothalamic releasing factors. We describe differential regulation of natriuretic peptide genes by the TRH, cAMP, and oestrogen signalling pathways, and identify putative CNP target genes in GH3 somatolactotropes. Furthermore, we provide the first description of a natriuretic peptide system in feline pituitaries.

## 2. Results

### 2.1. Molecular Characterization of a Custom Multiplex RT-qPCR Assay for Natriuretic Peptides and Somatotrope Markers in GH3 Somatolactotropes and Primary Rat Pituitary Tissue 

We have previously characterised the molecular and pharmacological properties of the natriuretic peptide system in GH3 cells [12] using conventional end-point RT-PCR. However, a comprehensive molecular analyses of the natriuretic peptide genes, their receptors, and other somatotrope-enriched genes has not previously been performed. Using the GeXP Fragment Analyser, we designed and optimised a custom multiplex RT-qPCR assay for *Nppa*, *Nppb*, *Nppc*, *Npr1*, *Npr2*, and *Npr3*, and examined the profile of these transcripts in GH3 cells. Specific transcripts were detected for each gene product included in the multiplex. As expected, GH3 cells expressed *Nppa*, *Nppb*, *Nppc*, *Npr1*, *Npr2,* and *Npr3* (Figure 1A).

### 2.2. cAMP and TRH Signalling Pathways Differentially Regulate Expression of Natriuretic Peptides and Their Receptors

We next examined whether the gene expression of the natriuretic peptides and their receptors could be regulated by the activation of signalling pathways known to be involved in somatotrope function. GH3 cells were stimulated for up to 24 h without or with 10 µM forskolin (FSK) or 100 nM thyrotropin releasing hormone (TRH), prior to RNA extraction and subsequent analyses by multiplex RT-qPCR. As shown (Figure 1B), *Nppc* expression increased in response to FSK stimulation (by 2.9 ± 0.2-fold ****, 2.7 ± 0.2-fold ***, and 2.1 ± 0.3-fold ** compared with untreated, at 4 h, 8 h, and 24 h, respectively; **** *p* < 0.0001, *** *p* < 0.001, ** *p* < 0.01). Similarly, *Nppc*, but not *Nppa* or *Nppb*, expression was up-regulated in response to TRH at 24 h (by 1.9 ± 0.2-fold compared with untreated, ** *p* < 0.01; see Figure 1C). In contrast, the expression of the natriuretic peptide receptors was less affected, with only *Npr1* expression being moderately increased in the presence of FSK at 24 h (by 1.5 ± 0.2-fold, * *p* < 0.05; see Figure 1D,E). To determine the mechanism by which FSK and TRH exerted their effects on *Nppc* expression, GH3 cells were pre-treated with the selective PKA inhibitor, H89 (10 µM), or the selective MEK inhibitor, PD98059 (1 µM) for 30 min, prior to stimulation with FSK or TRH. As shown (Figure 2A), FSK again caused an increase in *Nppc* expression, and this was significantly inhibited in the presence of the PKA inhibitor, H89 (2.6 ± 0.2-fold compared with 1.4 ± 0.05-fold, ** *p* < 0.01). Furthermore, TRH increased *Nppc* expression (Figure 2B), and PD98059 pre-treatment significantly inhibited this response (1.9 ± 0.2-fold compared with 1.03 ± 0.1-fold, * *p* < 0.05). Collectively, these data suggest that *Nppc* mRNA expression is regulated by PKA and MEK–ERK signalling pathways in GH3 cells.

### 2.3. Identification of Putative CNP Target Genes in GH3 Somatolactotropes

Patients with mutations in either the *NPPC* or *NPR2* genes frequently present with severe growth phenotypes [28,29,30]. Although much is known about the bone-specific influence of CNP action on growth characteristics [31,32], there is a paucity of data describing the potential biological role for CNP at the level of the somatotrope cell. Having shown that CNP expression (*Nppc*) could be regulated in GH3 cells by pathways used by activators of somatotrope function, we examined the reciprocal relationship by investigating the potential effects of CNP on somatotrope gene expression. We optimised a custom multiplex RT-qPCR assay that included primers specific to the following genes: *Insr, Lepr, Esr1, Trhr*, *Sstr1*, *Sstr2*, *Sstr3*, *Sstr4*, *Sstr5*, *Aip*, *cFos*, *Gh1*, *Pou1f1*, and *Prl*. As shown (Figure 3A), transcripts for each of these genes were detected in RNA extracted from untreated GH3 cells, with an expression pattern comparable to primary rat pituitary tissue. We then performed a series of time-course experiments, to determine the potential effect of CNP on gene expression. GH3 cells were serum-starved overnight, prior to stimulation with 100 nM CNP for up to 24 h, prior to the extraction of total RNA and analysis by multiplex RT-qPCR. The expression levels of *Gh1*, *Prl*, *Aip*, *Egr1,* and *Pou1f1* were not significantly altered in response to CNP (Figure 3B–E,G). However, the immediate early gene, *cFos*, showed increased expression following CNP stimulation for 4 h and 8 h (by 3.6 ± 0.5-fold and 3.4 ± 0.03-fold, respectively; ** *p* < 0.01; see Figure 3F). Interestingly, over the same time period, the expression of *Esr1*, *Insr,* and *Lepr* were each significantly inhibited within 4 h of CNP treatment (to 0.74 ± 0.08-fold (* *p* < 0.05), 0.7 ± 0.03-fold (** *p* < 0.01), and 0.4 ± 0.002-fold (**** *p* < 0.0001), respectively; see Figure 3G–I). In contrast, the expression of the G-protein coupled receptor panel (*Trhr*, *Sstr1*, *Sstr2*, *Sstr3*, *Sstr4*, *Sstr5*) were not significantly affected by CNP (Figure 3J,K). In summary, these data suggest a potential role for CNP in the regulation of *cFos, Esr1*, *Insr,* and *Lepr* expression in GH3 somatolactotropes.

### 2.4. Interaction of Oestrogen and Natriuretic Peptides in GH3 Somatolactotropes

Our initial experiments had indicated a role for cAMP and MEK–ERK signalling pathways in regulating the natriuretic peptide system in GH3 somatolactotropes. However, in addition to peptide hormone activation of these signalling pathways, somatolactotropes are also sensitive to changes in oestrogen concentrations in vivo [33,34]. Oestrogen has previously been shown to regulate the natriuretic peptide system in other tissues [35,36,37,38]. To establish whether a functional relationship existed between these pathways, GH3 cells were treated with 0 or 100 nM 17β-oestradiol (E2) for up to 24 h before extracting total RNA and performing RT-qPCR for natriuretic peptide genes. As shown (Figure 4A), E2 increased expression of *Nppa* (after 24 h) and *Nppc* (after 8 h) (to 2.2 ± 0.3-fold and 1.6 ± 0.3-fold compared with control, for *Nppa* and *Nppc*, respectively; * *p* < 0.05). Furthermore, expression of both *Npr1* and *Npr2* transcripts were increased following exposure to E2 for 8 h (Figure 4B; to 1.75 ± 0.2-fold and 2.5 ± 0.4-fold compared with control, for *Npr1* and *Npr2*, respectively; ** *p* < 0.01). 

Next, we investigated whether natriuretic peptide function in vitro was sensitive to oestrogen. GH3 cells were pre-treated with 0 or 100 nM E2 for either 24 h or 72 h, and then stimulated with 0 or 100 nM CNP in the presence of 1 mM IBMX, for a further 1 h. As shown (Figure 4C,D), CNP caused the expected increase in cGMP accumulation compared with control (from 3.8 ± 0.5 pmol/mL to 82.1 ± 19.4 pmol/mL, **** *p* < 0.0001; from 3.6 ± 1.2 pmol/mL to 55.0 ± 3.2 pmol/mL, **** *p* < 0.0001; 24 h and 72 h pre-treatment, respectively). However, pre-treatment with 100 nM E2 (at both pre-treatment time-points) caused a significant inhibition of CNP-stimulated cGMP accumulation (from 82.1 ± 19.4 pmol/mL to 36.5 ± 8.0 pmol/mL, * *p* < 0.05; from 55.0 ± 3.2 pmol/mL to 29.0 ± 5.0 pmol/mL, **** *p* < 0.0001; 24 h and 72 h pre-treatment, respectively). These data show that expression and function of the natriuretic peptide system in GH3 somatolactotropes are sensitive to regulation by oestrogen.

Finally, to investigate the potential for oestrogen to impact pituitary CNP in vivo, we measured pituitary CNP content throughout the oestrous cycle in rats. Vaginal smears were performed in order to characterise the stage of the oestrous cycle (Figure 4E), before the animals were euthanized on the morning of each stage of the cycle, followed by trunk blood collection and pituitary protein extraction. Radioimmunoassay for CNP showed that CNP content was unaffected by stage of the oestrous cycle; the lowest pituitary CNP content was found at proestrous, but this did not attain significance (Figure 4E). As expected, pituitary and plasma LH content were elevated at oestrous compared to diestrous and proestrous (to 2.42 ± 0.4-fold ** *p* = 0.0034 and 2.7 ± 0.5-fold ** *p* = 0.0017, for pituitary and plasma measurements, respectively). Overall, these data show that pituitary CNP content does not change appreciably throughout the oestrous cycle in rats.

### 2.5. Natriuretic Peptide System in Feline Somatotrope Tumours

GH3 cells represent a mixed lineage of somatotropes and lactotropes, having arisen from pituitary tumours in Wistar-Furth rats [39,40]. Similar to these models of rat pituitary tumours, we have previously shown that human somatotropinoma express both *NPPC* and *NPR2* [9]. To expand these studies, we took advantage of a spontaneously occurring companion animal model of acromegaly, and recruited a cohort of cats (*Felis catus*), diagnosed with hypersomatotropism (HST; acromegaly). Patient data and clinical signalment for this cohort were previously reported in [15]. Following surgical hypophysectomy, total RNA was extracted from the tissue, and feline-specific multiplex RT-qPCR assays were used to analyse expression of the natriuretic peptide system in feline HST tumours, as well as in normal feline pituitary tissue. As shown (Figure 5A), all three natriuretic peptides, and their receptors, were expressed in normal feline pituitary tissue. Comparing the expression of *NPPA*, *NPPB,* and *NPPC* in control (*n* = 10) and HST (*n* = 19) tissue revealed no significant differences in the expression of any of the three natriuretic peptide genes (Figure 5B–D). However, although neither *NPR2* nor *NPR3* mRNA expression were altered between healthy control cats and cats with HST (Figure 5F,G), *NPR1* expression was significantly increased in the pituitaries of cats with HST (from 0.15 ± 0.03 cf 0.40 ± 0.04, *** *p* = 0.0007; Figure 5E). Collectively, these data suggest that pituitary *NPR1* expression is enhanced in patients with HST, but the expression of other natriuretic peptides/receptor is unaffected.

Having demonstrated the presence of transcripts for all three natriuretic peptides and natriuretic peptide receptors in feline pituitary samples, we next examined potential relationships between the expressions of these genes with other somatotrope-expressed transcripts, reported in [15]. No significant correlations between natriuretic peptide transcripts and somatotrope-expressed genes were observed in pituitary samples from control cats. However, as shown, (Figure 6A–F), *NPPC* expression was positively correlated with *NPPA* (r = 0.55, * *p* = 0.015), *NPR2* (r = 0.5, * *p* < 0.05), *NPR3* (r = 0.83, ** *p* < 0.01), *D2R* (r = 0.66, ** *p* < 0.01), and *GHR* (r = 0.58, ** *p* < 0.01), and negatively correlated with *SSTR5* (r = −0.52, * *p* = 0.02). As CNP has been implicated in the negative control of proliferation in several tissues [41,42], we examined whether expression of any of the natriuretic peptides correlated with the size of the tumour in these patients. Transverse CT scans (Figure 6G) were examined from each patient, and dorsoventral (DV) mass of tumour calculated. Notably, there was a significant negative correlation between DV size and *NPPC* expression (Figure 6G; r = −0.54, *p* = 0.02), which was not seen with either *NPPA* or *NPPB*. These data suggest a potential interaction between CNP and regulators of somatotrope function in feline patients with HST. 

## 3. Materials and Methods

### 3.1. Materials

TRH, 17β-Oestradiol (E2), forskolin (FSK) and CNP were obtained from Sigma Aldrich (Poole, Dorset, UK). PD98059 (MEK inhibitor) was purchased from Calbiochem (Merck Life Science UK Limited, Watford, UK). H89 (PKA inhibitor) and 8-Br-cGMP were obtained from Tocris (Bio-Techne, Abingdon, UK). All other chemicals were purchased from Sigma, unless stated otherwise.

### 3.2. Cell Culture and cGMP Enzyme Immunoassay

GH3 somatolactotropes were grown as an adherent monolayer in DMEM supplemented with 10% (*v*/*v*) FCS and 1% (*v*/*v*) antimycotic/antibiotic, as described previously [12]. The cells were passaged twice weekly and cultured in an humidified incubator at 37 °C with 5% (*v*/*v*) CO_2_. For RNA experiments, cells were plated at 2.5 × 10^5^ cells/well in 12-well plates, allowed to adhere and serum starved for 4 h prior to treatment with DMEM supplemented with 1% (*v*/*v*) antimycotic/antibiotic. Cells were treated with either 100 nM TRH, 10 µM forskolin (FSK), or 100 nM CNP for up to 24 h. For experiments using pharmacological inhibitors, the cells were pre-treated with 0, 10 µM PD98059, or 1 µM H89 for 30 min, after which the cells were treated in the continued absence or presence of the inhibitors and either 0, 100 nM TRH, or 10 µM FSK. To investigate cGMP accumulation, GH3 cells were plated at 2.5 × 10^5^ cells/well in 12-well plates, allowed to adhere and switched to 10% (*v*/*v*) charcoal-stripped FCS and 1% (*v*/*v*) antimycotic/antibiotic, in the absence or presence of 0 or 100 nM 17β-Oestradiol for up to 72 h, before being stimulated in physiological saline solution (PSS; 127 mM NaCl, 1.8 mM CaCl_2_, 5 mM KCl, 2 mM MgCl_2_, 0.5 nM NaH_2_PO_4_, 5 mM NaHCO_3_, 10 mM glucose, 0.1 % (*w*/*v*) BSA, 10 mM HEPES, adjusted to pH 7.4) with 1 mM 3-isobutyl-1-methylxanthine (IBMX), and either 0 or 100 nM CNP for 1 h. Stimulations were terminated with ice-cold 100% (*v*/*v*) ethanol, and samples were dried down under vacuum, before being assayed using a cGMP-enzyme immunoassay (R&D Systems, Abingdon, UK) as described previously [12].

### 3.3. Patient Data, Ethics, and Hypophysectomy Surgery

Ethical approval for this study was obtained from the Royal Veterinary College (RVC) Ethics and Welfare Committee (URN 2014 1306). All feline pituitary samples (patient (*n* = 19, 18 male) and controls (*n* = 10, 7 male)) have been previously described in [15]; all cats recruited to the study were neutered. Briefly, pituitary tumour samples were obtained from cats diagnosed with hypersomatotropism (HST, Acromegaly) according to appropriate clinical history, a serum IGF1 concentration of >1000 ng/mL, and intracranial imaging of a pituitary mass. Written informed consent was received from the owners of all the cats enrolled on the study. Control pituitary tissue was collected at post-mortem examination. All pituitary tissue used in this study (patient and controls) was stored in RNAlater™ (Sigma-Aldrich, Poole, Dorset, UK) prior to RNA extraction. 

### 3.4. RNA Extraction

Total RNA was extracted from GH3 cells, primary rat pituitary, or feline pituitary biopsies using RNAbee (AMS Biotechnology, Abingdon, Oxford, UK) as previously described [12], and subjected to DNase treatment using Qiagen RNeasy kit, in accordance to manufacturer’s instructions. RNA concentrations were determined using ND-100 spectrophotometer (NanoDrop^TM^), and 260/280 ratios were used to maintain RNA integrity. RNA samples were also routinely inspected using the Agilent Bioanalyzer (Agilent Technologies, Cheadle, Cheshire, UK), with typical RIN values of >8 being obtained.

### 3.5. GeXP Multiplex RT-qPCR

Customised GeXP multiplex assays were designed to genes that included the natriuretic peptide system as well as somatotrope markers (see Appendix A). Target-specific reverse transcription (using 100 ng RNA as template), and PCR amplification were performed as we described previously [15,43,44], and in accordance with manufacturer’s instructions (Beckman Coulter, High Wycombe, UK). In brief, a master mix was prepared for reverse transcription as detailed in the GeXP starter kit (Beckman Coulter, High Wycombe, UK) and performed using a G-storm GS1 thermal cycler (Agilegene Technologies Ltd., Somerton, Somerset, UK), using the following protocol: 48 °C 1 min; 42 °C 60 min; and 95 °C 5 min. From this an aliquot of each reverse transcriptase reaction was added to PCR master mix, consisting of GenomeLab kit PCR reaction mix and Thermo Scientific Thermo-Start Taq DNA polymerase. PCR reactions were performed using G-Storm GS1 thermal cycler with a 95 °C activation step for 10 min, followed by 35 cycles of 94 °C 30 s; 55 °C 30 s; 70 °C 60 s. Products were separated and quantified using CEQTM 8000 Genetic Analysis System, and GenomeLab Fragment Analysis software (Beckman Coulter, High Wycombe, UK).

### 3.6. Immunoassays and Vaginal Cytology

Vaginal cytology was performed on random cycling female Sprague–Dawley rats between the ages of 8 to 10 weeks, to determine oestrous cycle staging, as described previously [45]. Rats were euthanised on the morning of each stage of the cycle by cervical dislocation, and pituitaries were removed. Trunk bloods were collected for plasma analysis of LH. Pituitary samples were sonicated, and the resulting suspension centrifuged at 450× *g* and 4 °C for 5 min. Supernatants were dried under and resuspended with assay buffer before CNP content was determined using a commercially available RIA (Peninsula Laboratories, Wirral, UK). Both pituitary and plasma LH content was measured using RIA reagents kindly provided by the NIDDK (Bethesda, MD, USA), as described previously [46]. 

### 3.7. Data Presentation and Statistical Analyses

Data shown are means ± SEM from individual RNA extractions, pooled from experiments performed in duplicate or triplicate. GH3 cells were used from a wide range of passages (from 8 to 35). Numerical data were subjected to ANOVA, followed by Tukey’s or Dunnett’s multiple comparison tests (where appropriate), or correlations using Spearman’s non-parametric test (accepting *p* < 0.05), using in-built equations in GraphPad Prism 8.0 for Mac (GraphPad, San Diego, CA, USA). Immunoassay data were also analysed using in-built sigmoidal nonlinear curve-fitting equations in GraphPad Prism 9.0 for Mac.

## 4. Discussion

Since its discovery, the initial expression profiling descriptions of CNP in the pituitary have been linked with the control of growth and reproduction [1,47,48,49], although direct functional effects of CNP on these endocrine axes remain poorly described. Several studies reinforced the position of CNP as the major natriuretic peptide of the anterior pituitary [9,10,11,44]. Despite these findings, understanding of the regulation and function of CNP and GC-B function in the pituitary is limited. In this study, we provide further evidence for a regulated natriuretic peptide system in pituitary tumour-derived GH3 cells as well as in feline patients with hypersomatotropism (acromegaly) due to pituitary tumours.

Although CNP performs a broader range of biological functions throughout the CNS and peripheral tissues compared with the predominantly cardiovascular roles for ANP and BNP, it is the effects of CNP on endocrine ossification that are best described. Numerous loss-of-function mutations have now been reported in *NPPC* and *NPR2*, leading to profound skeletal dysplasias and short stature [8,28,50]. Mouse models of disrupted *Nppc* and *Npr2* essentially phenocopy the clinical signs observed in humans, but in addition, some of these mouse models have reduced pituitary GH, suggesting a positive role for CNP in pituitary GH secretion [5,6]. Our previous investigations established that a complete natriuretic peptide system (peptides, receptors, processing enzymes) is expressed in both rodent pituitaries [11], normal human pituitary tissue (adult and fetal), as well as human pituitary tumours of different developmental origins [9]. Here, we provide evidence of a natriuretic peptide system in rat GH3 somatolactotropes and feline pituitary tumours from patients with HST. Furthermore, our data suggest enhanced expression of *NPR1* in feline pituitary tumour samples compared with normal feline pituitary tissue, which might indicate a potential regulatory role for ANP/NPR1 signalling in feline pituitary function. However, our expression data from mouse and rat pituitary tissue and cell lines support a potential role for CNP/NPR2, rather than ANP/NPR-1, in local regulation of pituitary function [10,11,12,51]. Moreover, mouse models of *Nppa/Npr1* disruption do not phenocopy the growth-related phenotypes of *Nppc/Npr2* models [52,53]. Nevertheless, our correlation analyses of natriuretic peptide gene expression in feline pituitary tumour samples reveal relationships between the different peptides and their receptors, as well as genes encoding physiological regulators of somatotrope function (*D2R, SSTR5,* and *GHR*). In contrast, we found no such correlations between natriuretic peptide gene transcripts and *D2R, SSTR5,* and *GHR* in pituitaries from control cats. This likely reflects the enriched expression of these physiological regulators within tumour cells of somatotrope-origin, compared with their expression across a range of different cell types in normal pituitary samples. 

In addition to these relationships between natriuretic peptides and somatotrope regulators in feline HST tumours, we found that pituitary *NPPC* expression displayed a modest negative correlation with tumour volume, suggesting there may be a role for CNP in controlling proliferation. Such an inhibitory effect is at odds with previous studies suggesting that CNP and cGMP positively regulate GH secretion [22,23]. It may well be that activation of multiple signalling pathways downstream of the GC-B receptor could influence proliferation, and we have recently shown CNP-stimulated ERK1/2 activation to occur at concentrations several orders of magnitude below that required for cGMP generation [51]. An important caveat in interpreting our gene expression data from feline pituitary tumour samples is that surgical treatment of these patients results in the removal of the entire pituitary (hypophysectomy), rather than resection of tumour tissue [54]; as a result, these samples might contain some normal pituitary tissue as well as pituitary tumour tissue. Despite these limitations, the current data represent the first description of a pituitary natriuretic peptide system in normal feline pituitary tissue and feline pituitary tumours, which could inform future therapeutic approaches. 

Endocrine regulation of the expression of natriuretic peptide genes has previously been reported in a range of tissues and cell lines [41,42,44,55,56], although the mechanisms behind this regulation are still unclear. Our current study suggests the involvement of the cAMP/PKA and MEK–ERK signalling pathways in the positive regulation of endogenous *Nppc* expression in GH3 somatolactotropes. The initial description of the *Nppc* promoter included the identification of a single cAMP response element (CRE) within a very proximal part of the murine promoter [57], which may explain the effects of forskolin described here. However, the transcription factor most closely associated with transcriptional effects mediated via the CRE, CREB (cAMP response element binding protein), is subject to post-translational modification on Ser133 by numerous protein kinases, including PKA and ERK1/2 [58,59]. Therefore, CREB regulation of *Nppc* transcription might serve as an integrator of multiple signalling pathways. cAMP has previously been shown to enhance *Nppc* expression in the ovine pars tuberalis [60], and in rat FRTL-5 thyroid follicular cells [61]. The effect of TRH on *Nppc* expression in GH3 cells was shown to involve ERK1/2 signalling, which is consistent with what is known about TRH signalling in these cells [62]. The TRH effect on *Nppc* expression is of similar magnitude to that seen in GnRH-stimulated, gonadotrope-derived LβT2 cells [44], a cell line that is known to show robust ERK1/2 responses to GnRH [63,64]. In our previous studies, we did not observe CREB-associated proteins forming complexes with the proximal *Nppc* promoter in vitro, but did identify binding of both Sp1 and Sp3 [11]. Sp1, in particular, is known to be cAMP responsive in gonadal tissue, where it mediates the LH response on *Klf4* expression [65], and in human granulosa cells where FSH regulated Sp1-dependent *IRS2* expression [66]. In our current study, of the three natriuretic peptide receptors, only *Npr1* expression was significantly altered by forskolin, after 24 h treatment. In common with *Nppc* expression, transcriptional regulation of *Npr1* also involves Sp1 and Sp3 [67]. Whilst our data do not provide a precise mechanism by which *Nppc* and *Npr1* expression is altered, it is likely that the effects of forskolin and TRH on *Nppc* and *Npr1* expression in GH3 somatolactotropes involves transcriptional regulation via these Sp1 sites.

An increasing number of studies have reported putative target genes downstream of CNP signalling [68,69], yet potential CNP target genes in the pituitary remain poorly defined. In the gonadotrope-derived LβT2 cell line, CNP caused an increase in the expression of *cJun*, *Egr1*, *Nr5a1,* and *Nr0b1*, within 8 h of stimulation [44], but failed to alter *cFos* expression. In contrast, the present study revealed a robust *cFos* response to CNP (within 4 h) in GH3 cells, but no significant effects on *Egr1*, suggesting that cell-type specific effects of CNP may occur. The rapid induction of *cFos* expression observed here is in keeping with the observed effects of CNP in regulating *cFos* in PC12 pheochromocytoma and C6 glioma cells [68]. Surprisingly, *Gh1* expression was not significantly altered by CNP treatment in GH3 cells. Previous studies have shown that both CNP and cGMP can act as GH secretagogues in GH3 cells and rat pituitary cells [22,23]. The lack of effect on *Gh1* expression in our current studies suggests that CNP may only regulate GH secretion. Of the other somatotrope-enriched transcripts included in our multiplex RT-qPCR assay, only *Esr1*, *Insr,* and *Lepr* expression were inhibited by CNP. Somatotropes have long since been identified as oestrogen target cells, and have well-documented effects on regulating growth [70]. More recently, somatotropes have been recognised as metabolic sensors, responding to circulating insulin [71] and leptin [72,73]. Our finding that CNP can inhibit expression of the receptors for oestrogen, leptin and insulin, suggests a potential role for fine-tuning the somatotrope response to these metabolic signals. However, the exact mechanism by which CNP is exerting these changes in gene expression in GH3 cells remains to be determined. The effects of cGMP on gene expression can involve multiple potential pathways, including regulation of PKG and PKA signalling [74], leading to changes in gene transcription or RNA stability [75,76]. As cFos is a component of the AP-1 transcriptional heterodimer [77], it is tempting to speculate that some of these effects of CNP are mediated via AP-1 sites in target gene promoters.

Gonadal steroids have previously been shown to regulate the expression of natriuretic peptide genes in a range of tissues [39,40,41,42,55]. 17β-oestradiol is known to upregulate *Nppa* expression in cardiomyocytes [39], whereas uterine expression of both ANP and CNP are altered throughout the oestrous cycle [42]. Our data reveal that *Nppa*, *Nppc*, *Npr1,* and *Npr2* expression is enhanced following stimulation with 17β-oestradiol in GH3 somatolactotropes. In contrast, functional analyses of CNP-stimulated cGMP accumulation in GH3 cells following exposure to 17β-oestradiol for between 24 h and 72 h revealed a significant reduction in cGMP levels, which may represent a pharmacological mechanism to control the local effects of CNP in the pituitary. However, our in vivo studies showed that despite confirming a robust change in pituitary and plasma LH concentrations throughout the oestrous cycle, pituitary CNP content did not change significantly. The apparent discrepancy between positive changes to *Nppc* mRNA expression in GH3 cells, and the lack of significant changes in pituitary CNP protein in rats might reflect the contribution of other cell types in primary pituitary tissue. In addition, these inhibitory effects of oestradiol on CNP action in GH3 somatolactotropes is also at odds with the positive effects seen in the ovary [55]. A potential confounding issue is the timing of these oestrous cycle studies. Pituitaries and blood samples were collected on the morning of each stage of the oestrous cycle; as a result, the proestrus surge in LH had yet to occur. There remains a need for comprehensive mechanistic studies in order to establish the relationship between oestrogen and CNP signalling in the anterior pituitary. 

The systemic effects of CNP on growth are well characterised, with several CNP analogues now in clinical trials for patients with short stature [24,25,26,27,78]. Despite these bone-mediated effects of CNP, our molecular studies identify several potential CNP targets in somatotrope cells that may influence the growth hormone axis. Further studies are required to better characterise how CNP might regulate somatotrope gene expression and function, in normal pituitary tissue as well as in patients with acromegaly.

## Figures and Tables

**Figure 1 ijms-22-01076-f001:**
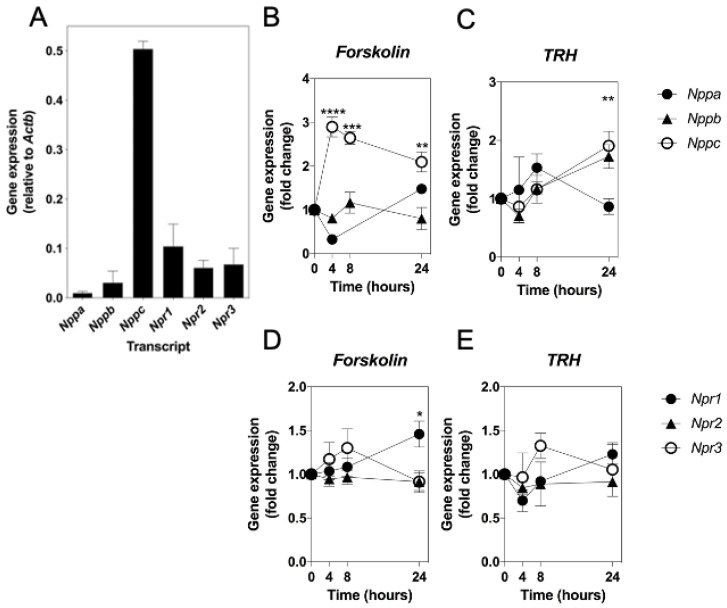
Molecular characterisation and regulation of the natriuretic peptide system in GH3 somatolactotropes. (**A**) Expression of natriuretic peptide and natriuretic peptide receptor genes in GH3 cells. Data shown are means ± SEM of relative gene expression (normalised to *Actb*) from 3 to 8 individual RNA extractions. (**B**–**E**) Effect of forskolin (FSK) and TRH on natriuretic peptide and natriuretic peptide receptor gene expression in GH3 cells. Cells were treated for up to 24 h with either 0, 10 µM FSK (**C**,**E**), or 100 nM TRH (**D**) before extraction of RNA, prior to multiplex RT-qPCR assay. Data shown are means ± SEM normalised as fold changes compared to 0 time-point, from 3 to 5 individual RNA extractions (**** *p* < 0.0001, *** *p* < 0.001, ** *p* < 0.01, * *p* < 0.05, significantly different from 0 time-point).

**Figure 2 ijms-22-01076-f002:**
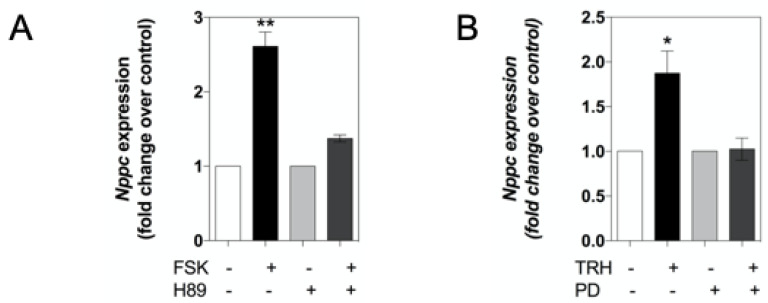
Role of PKA and MEK–ERK signalling pathways in the regulation of *Nppc* expression. Effect of (**A**) PKA inhibitor (H89) on FSK-stimulated, or (**B**) MEK inhibitor (PD98059) on TRH-stimulated, *Nppc* gene expression in GH3 cells. Cells were pre-treated for 30 min with either 0, 10 µM H89 or 1 µM PD98059 before stimulation with either 0, 10 µM FSK or 100 nM TRH for 24 h, followed by extraction of RNA and subsequent multiplex RT-qPCR assay. Data shown are means ± SEM normalised as fold changes compared to control, from 3 to 6 individual RNA extractions. (** *p* < 0.01, * *p* < 0.05, significantly different from control).

**Figure 3 ijms-22-01076-f003:**
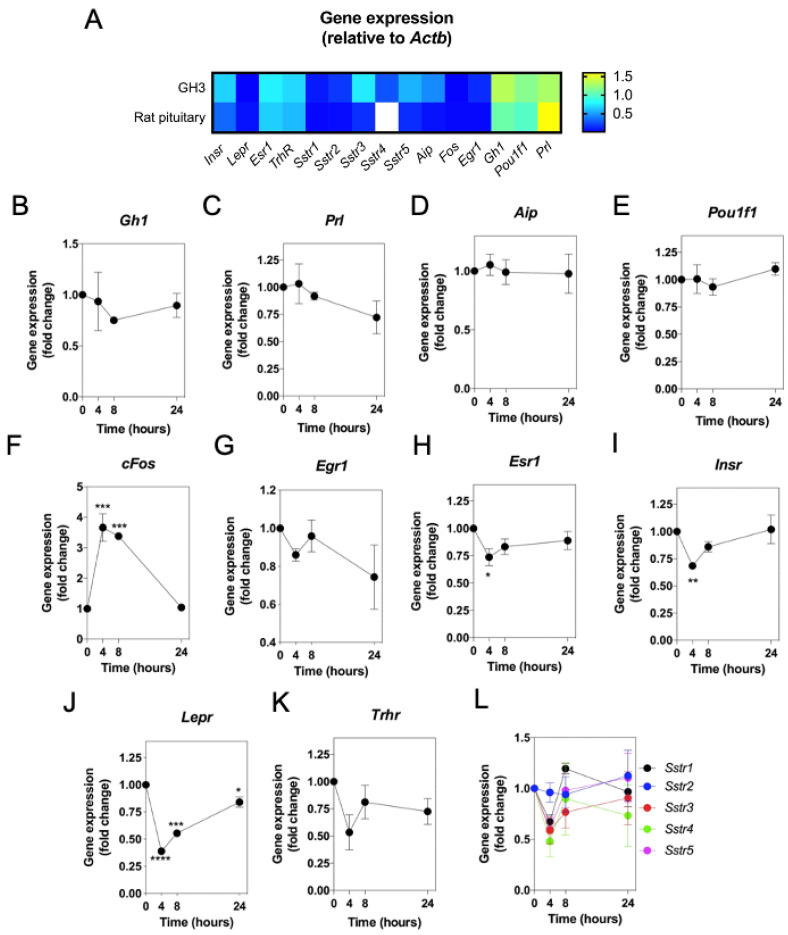
Effect of C-type natriuretic peptide (CNP) on somatolactotrope-enriched gene expression in GH3 cells. (**A**) Heatmap of relative gene expression in rat pituitary and GH3 somatolactotropes (normalised to *Actb*). (**B**–**L**) GH3 cells were cultured in the presence of 0 or 100 nM CNP for up to 24 h, before extracting RNA and performing multiplex RT-qPCR to examine alterations in gene expression profiling of somatolactotrope-enriched transcripts (*Insr*, *Lepr*, *Esr1*, *Trhr*, *Sstr1*, *Sstr2*, *Sstr3*, *Sstr4*, *Sstr5*, *cFos*, *Egr1*, *Gh1*, *Prl*, *Aip*, *Pou1f1*, *Fos*). Data shown are means ± SEM (*n* = 3 to 5 individual RNA extractions) of relative gene expression (normalized to *ActB*); (**** *p* < 0.0001, *** *p* < 0.001, ** *p* < 0.01, * *p* < 0.05, significantly different from basal.

**Figure 4 ijms-22-01076-f004:**
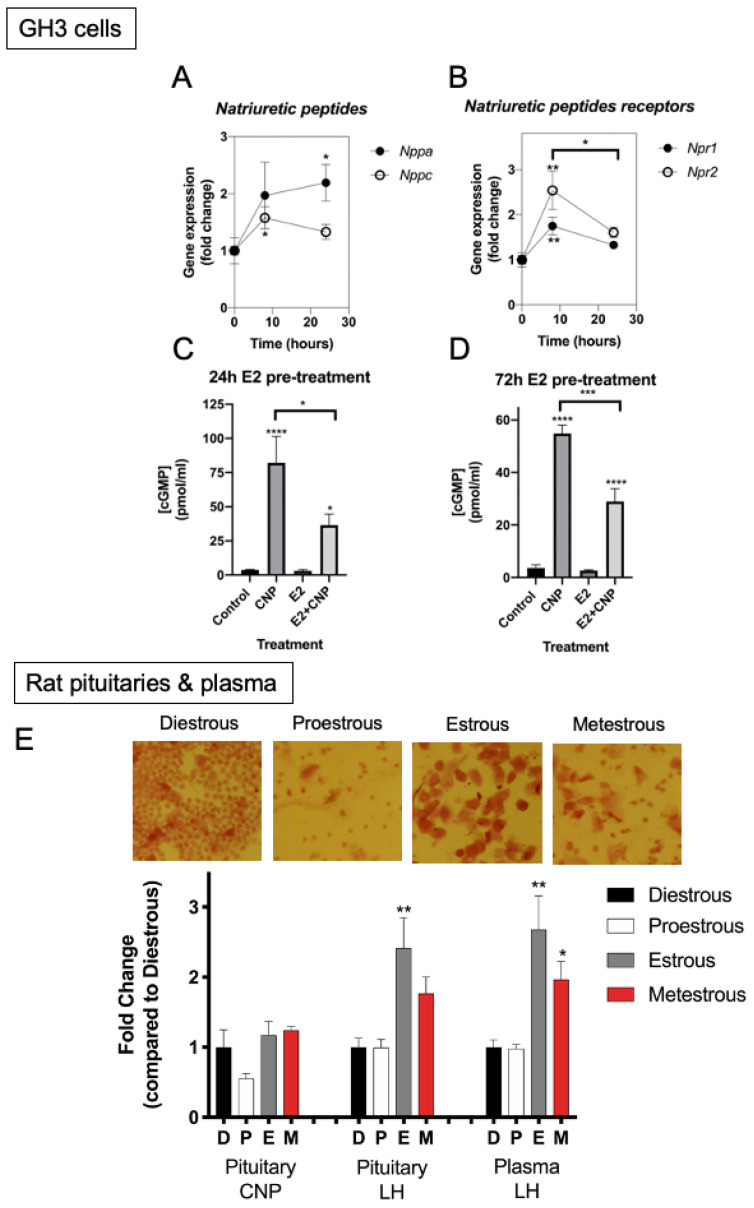
Effect of 17β-oestradiol on natriuretic peptide expression and function in GH3 somatolactotropes and rat anterior pituitaries. (**A**,**B**) GH3 cells were cultured in the presence of 0 or 100 nM 17β-oestradiol for up to 24 h. RNA was extracted and analysed by multiplex RT-qPCR to examine alterations in gene expression profiling of natriuretic peptide system transcripts (*Nppa*, *Nppc*, *Npr1*, *Npr2*). Data shown are means ± SEM (*n* = 3 to 5 individual RNA extractions) of relative gene expression (normalized to *ActB*); * *p* < 0.05, ** *p* < 0.01, significantly different from basal. (**C**,**D**) GH3 cells were cultured in the presence of 0 or 100 nM 17β-oestradiol for 24 h and 72 h, followed by exposure to 0 or 100 nM CNP in the presence of 1 mM IBMX for 1 h. Total cGMP concentration were determined using a commercially available enzyme immunoassay kit. Data shown are means ± SEM of 5 independent experiments, each performed in triplicates; **** *p* < 0.0001, *** *p* < 0.001, * *p* < 0.05, significantly different from control, or in the presence of E2 (as indicated). (**E**) Effect of oestrous cycle on pituitary CNP content. Vaginal smears were performed on random cycling Sprague–Dawley rats (8–10 weeks old); many neutrophils were present at diestrous, with small nucleated epithelial cells present at proestrous, and predominantly keratinized epithelial cells at estrous and metestrous. Pituitary CNP and LH content, and plasma LH were measured by RIAs. Data shown are means ± SEM from 3 to 12 animals, normalised as fold changes compared to diestrous (** *p* < 0.01, * *p* < 0.05, significantly different from diestrous).

**Figure 5 ijms-22-01076-f005:**
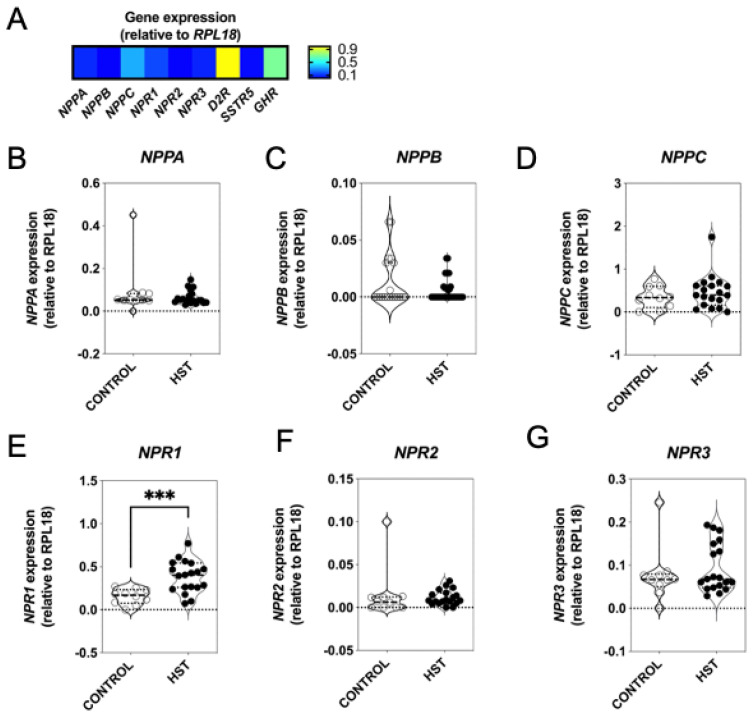
Molecular characterisation of the natriuretic peptide system in feline somatotrope tumours. (**A**) Optimisation of the multiplex RT-qPCR assay to detect natriuretic peptide transcripts and somatotrope-enriched genes from feline pituitary tissue, showing relative expression heatmap of transcripts, normalised to *RPL18*. (**B**–**G**) Total RNA was extracted from feline pituitary samples taken from patients with hypersomatotropism (HST) (*n* = 19) and healthy controls (*n* = 10), and analysed by multiplex RT-qPCR to examine alterations in gene expression profiling of the natriuretic peptide system (*NPPA*, *NPPB*, *NPPC*, *NPR1*, *NPR2*, *NPR3*). Violin plots show median and quartiles; *** *p* < 0.0001, significantly different compared to control.

**Figure 6 ijms-22-01076-f006:**
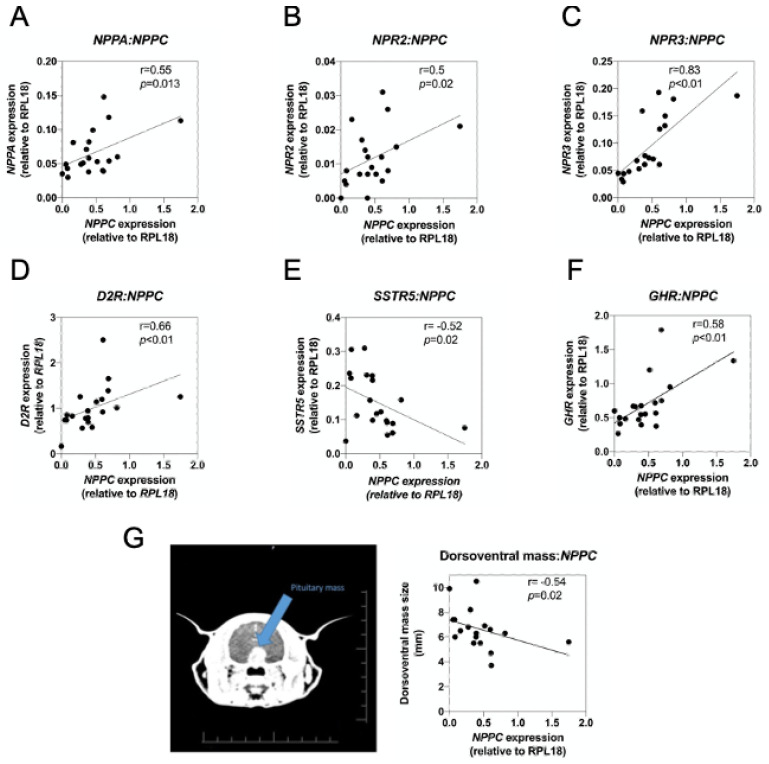
Correlation relationships between *NPPC* and somatotrope-enriched genes in feline somatotrope tumours. (**A**–**F**) Dot plots showing significant correlations between *NPPC* and (**A**) *NPPA*, (**B**) *NPR2*, (**C**) *NPR3*, (**D**) *D2R*, (**E**) *SSTR5,* and (**F**) *GHR*. (**G**) Representative transverse CT scan of feline pituitary tumour, from which dorsoventral mass size was measured (in mm), and correlated with *NPPC* expression. Data shown are from 19 feline HST patients.

## Data Availability

The data presented in this study are available on request from the corresponding author. The data are not publicly available due to patient privacy.

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
