# Peer review of "Natriuretic Peptide Expression and Function in GH3 Somatolactotropes and Feline Somatotrope Pituitary Tumours"

_ijms, 2021, doi:10.3390/ijms22031076_

Round 1
Reviewer 1 Report
The article “Natriuretic peptide expression and function in GH3 somatolactotropes and feline somatotrope pituitary tumors” explores expression of CNPs and their receptor GC-B in a pituitary cell line and examine the signaling pathways that impact their expression. They also determined the effect of CNP on key signaling molecules in the cell line. Finally, they show clinical relevance of this data by demonstrating feline pituitary tumors hypersecreting growth hormone possess the CNP/GC-B system and there are some interesting correlations with tumor volume and expression of other molecules. This manuscript has several strengths. It is a logical follow-up to the groups previously published work and the data are critical for understanding the function of the CNP system in growth hormone secretion in normal and pathological states. The experiments are clearly described, well controlled, and analyzed with the appropriate statistical rigor. The manuscript is exceeding well written. There are minor weaknesses, mainly centered on lack of protein data.
The following comments are offered
For the feline pituitary analysis, what was the sex of the control and patient samples?
Were there changes in proliferation of the GH3 cells in response to CNP treatment? Since you examined tumor size in the felines and saw a negative correlation between NPPC expression and tumor size, it raises this question.
It would be helpful to have some protein expression analysis to accompany the qPCR data. This may be most relevant for the estrogen experiments. Although Nppa and Nppc are both increased at the mRNA level by E2, is there an accompanying increase in overall CNP expression or cGMP? The lack of this data does not substantially detract from the overall conclusions, though.
For the flow of the story, it would potentially make more sense to have the feline data at the end. However, it certainly reads fine as is.
In the methods under statistics, the RIA analysis should be included.
Minor edits:
Line 74: the word on should be of
Line 147: the word rat should appear after Sprague Dawley.
Figure 1B, gene expression should not have brackets around it
Line 225 should have a right bracket around the references
Author Response
Reviewer 1
For the feline pituitary analysis, what was the sex of the control and patient samples?
Thank you for raising this – apologies for not including this information in the original manuscript. There were 19 HST patients (18 male), and 10 controls (7male) – this male predilection has been observed previously in cats. All cats recruited for this study had been neutered. We have added all of this information to the Materials & Methods (p3, lines 123-125).
Were there changes in proliferation of the GH3 cells in response to CNP treatment? Since you examined tumor size in the felines and saw a negative correlation between NPPC expression and tumor size, it raises this question.
We have previously examined the potential effects of CNP on GH3 cell proliferation (by cell counting, FACS, MTT assay and thymidine incorporation) and failed to see any significant effect (Jonas et al. 2017). We did not repeat these experiments in the current study. Of course, interpreting cell proliferation data performed in tumorigenic cell lines must be done with caution, and the lack of effect of CNP on proliferation in vitro does not necessarily mean CNP does not affect proliferation in vivo.
It would be helpful to have some protein expression analysis to accompany the qPCR data. This may be most relevant for the estrogen experiments. Although Nppa and Nppc are both increased at the mRNA level by E2, is there an accompanying increase in overall CNP expression or cGMP? The lack of this data does not substantially detract from the overall conclusions, though.
We agree with the reviewer, and ideally these data would be followed up with investigations at the protein level. However, this would represent a significant amount of additional experimentation, and we have not had access to our laboratories since March 2020. With regards to cGMP, in our current study GH3 cells exposed to estrogen between 24h and 72h prior to CNP stimulation showed reduced cGMP accumulation.
For the flow of the story, it would potentially make more sense to have the feline data at the end. However, it certainly reads fine as is.
Thank you for this suggestion; we agree and have altered the sequences and references, accordingly.
In the methods under statistics, the RIA analysis should be included.
Apologies for the omission – we have added the methodology for the cGMP-EIA and the assay analysis in the Materials and Methods (p3, lines 111-119; p4, lines 172-173)
We have addressed all the minor edits (below), thank you.
Line 74: the word on should be of
Line 147: the word rat should appear after Sprague Dawley.
Figure 1B, gene expression should not have brackets around it
Line 225 should have a right bracket around the references
Reviewer 2 Report
This is an interesting study looking at natriuretic peptide expression and function in somatotrope cells of the pituitary.
Comments:
- Nppc expression in GH3 cells at baseline and following stimulation with Forskolin/TRH is higher than other natriuretic peptides. This is an interesting finding, but it would be good to compare this response in healthy vs tumour somatotropes to understand if this is relevant to disease?
- You suggest that induction of Nppc gene expression by forskolin and TRH involves activation of PKA/ERK pathways. Have you considered the involvement of NPR-3 in this pathway? It appears that NPR-3 gene expression may increase at 8hr following stimulation with forskolin & TRH? CNP can exert biological effects via activation of NPR-3 which is a Gi coupled receptor that can inhibit AC and cAMP production – negative feedback in this system? NPR-3 stimulation by CNP is also known to alter ERK 1/2 phosphorylation. It may be interesting to look at this as NPR-3 knockout mice exhibit skeletal abnormalities including long limbs.
- One of the aims of the study was to examine the direct functional effect of CNP in pituitary cells, do you have any other measures of somatotrope function apart from gene expression? eg. can you directly measure hormone secretion?
- Figure 3A you show a heat map of the genes that you examined in GH3 cells following stimulation with CNP in primary rat pituitary. Did you look at Nppc and natriuretic peptide receptor expression in this tissue too?
- Is your control/housekeeping gene (Actb) expression affected by any of the treatments with Forskolin/TRH/CNP/Estrogen? Also, you do not describe how you performed the gene expression analysis in the methods – did you use the delta CT method?
- In the discussion line 382 ‘such an inhibitory effect is at odds with previous studies suggesting that CNP and cGMP positively regulate GH secretion. It may well be that activation of multiple signalling pathways downstream of the GC-B receptor could influence proliferation’ – perhaps NPR-3 (NPR-C) is playing a role here? CNP can exert opposing effects on proliferation in vascular cells eg. it has been shown to inhibit proliferation in smooth muscle cells but stimulate growth of endothelial cells via NPR-C (Khambata et al 2011).
Author Response
Reviewer 2
Nppc expression in GH3 cells at baseline and following stimulation with Forskolin/TRH is higher than other natriuretic peptides. This is an interesting finding, but it would be good to compare this response in healthy vs tumour somatotropes to understand if this is relevant to disease?
The chemistry for the multiplex RT-qPCR assay is adapted to permit the function of multiple different primer sets in the same reaction, under the same conditions. But it is not possible to guarantee that each primer set functions with the same efficiency. It is important to note that the chemistry does not allow for comparisons between different transcripts within the same sample – but it does allow for comparisons of the same transcripts between different samples. Therefore, we do not make any statements about relative abundance of the different transcripts. We have compared the expression of some of the transcripts included in our study, between GH3 cells and rat pituitary tissue – of relevance to the referee’s query, there is significantly more Nppc transcript in GH3 cells compared with primary rat pituitary tissue (relative gene expression: 0.50±0.02 vs 0.16±0.01, P<0.0001). Given that primary rat pituitary tissue represents a mixed population of cell types, compared to the relatively homogeneous GH3 cell population, it is difficult to interpret whether the difference in Nppc expression is because of this, or due a difference in Nppc expression in primary somatotropes.
You suggest that induction of Nppc gene expression by forskolin and TRH involves activation of PKA/ERK pathways. Have you considered the involvement of NPR-3 in this pathway? It appears that NPR-3 gene expression may increase at 8hr following stimulation with forskolin & TRH? CNP can exert biological effects via activation of NPR-3 which is a Gi coupled receptor that can inhibit AC and cAMP production – negative feedback in this system? NPR-3 stimulation by CNP is also known to alter ERK 1/2 phosphorylation. It may be interesting to look at this as NPR-3 knockout mice exhibit skeletal abnormalities including long limbs.
Thank you for these very interesting and relevant questions. Of the 3 receptor transcripts, Npr3 expression appears to alter in response to TRH, but failed to reach significance in our studies. Our previous studies (Thompson et al 2014; Jonas et al, 2017 – both cited in the original manuscript), struggled to detect Npr3 transcripts, but we were using conventional end point PCR (which is less sensitive than the multiplex RT-qPCR). In Jonas et al, 2017, we perform a functional assay to examine the possible G-alpha-i coupling of Npr3, and failed to see any inhibition of cAMP signalling. This study also describes ERK1/2 activation in response to CNP in GH3 cells, that are dependent upon GC-B isoforms (as confirmed through siRNA silencing of Npr2). Therefore, whilst we agree that the role of Npr3 should not be overlooked (as a fundamentally important regulator of CNP signaling), we do not have strong evidence that it is functionally important in GH3 cells.
One of the aims of the study was to examine the direct functional effect of CNP in pituitary cells, do you have any other measures of somatotrope function apart from gene expression? eg. can you directly measure hormone secretion?
We agree that other functional parameters would be important to investigate down-stream of CNP signaling, and hormone secretion (GH and PRL, in particular) would be relevant to measure from GH3 cells – we have yet to perform such studies. Unfortunately, we have not been given access to our laboratories since March 2020, and are unable to start any experimentation until that situation resolves. In this manuscript, we do look at cGMP signaling as a function of CNP activity in GH3 cells, and show that it is negatively regulated by estradiol; but we agree with the reviewer that further functional effects will be important to establish.
Figure 3A you show a heat map of the genes that you examined in GH3 cells following stimulation with CNP in primary rat pituitary. Did you look at Nppc and natriuretic peptide receptor expression in this tissue too?
Please see our response to query 1. In addition, we have also described the expression and function of natriuretic peptides and their receptors, using end-point PCR, cell proliferation and cGMP assays in our previous studies (Thompson et al., 2014; Thompson et al. 2009).
Is your control/housekeeping gene (Actb) expression affected by any of the treatments with Forskolin/TRH/CNP/Estrogen? Also, you do not describe how you performed the gene expression analysis in the methods – did you use the delta CT method?
We routinely run three housekeeping genes as part of our multiplex assays (where the algorithm for compatibility permits) – Rpl19 and Gapdh are normally included, as well as Actb. We have consistently found Actb to be the most stable of these three housekeeping genes, in our studies performed in anterior pituitary cell lines (and note Actb is frequently used by other groups). However, we have not performed any sort of GeNorm analysis in GH3 cells, and accept that no housekeeping gene is perfect – all gene expression data were normalised to Actb expression within the same sample (one of the many benefits of the multiplex approach). The multiplex RT-qPCR technology that we, and others, have employed takes advantage of fragment analysis to discriminate between transcripts of different size, measuring the fluorescent signal as a means of quantification. As such, this means multiple transcripts can be detected from a single sample, within the same PCR conditions; hence, there is no delta CT performed as part of the analysis. In our manuscript, we cite our previous studies (Scudder et al, 2018; Staines et al, 2016; Mirczuk et al, 2019) that have employed this multiplex technique.
In the discussion line 382 ‘such an inhibitory effect is at odds with previous studies suggesting that CNP and cGMP positively regulate GH secretion. It may well be that activation of multiple signalling pathways downstream of the GC-B receptor could influence proliferation’ – perhaps NPR-3 (NPR-C) is playing a role here? CNP can exert opposing effects on proliferation in vascular cells eg. it has been shown to inhibit proliferation in smooth muscle cells but stimulate growth of endothelial cells via NPR-C (Khambata et al 2011).
Thank you for this, it is an interesting point. Please see our response to query 2, where we mention the potential for ERK signalling (downstream of GC-B). We had already included the possibility of activation of conflicting signaling pathways in our original discussion.
Reviewer 3 Report
The manuscript: Natriuretic peptide expression and function in GH3 somatolactotropes and feline somatotrope pituitary tumors, follows the long line of investigation on the role of natriuretic peptides in the anterior pituitary cells. The manuscript brings novel knowledge on the subject and is very well written. There are only a few minor issues to be resolved.
Minor comments:
Line 72, reference 10. Should it be PMID: 7682940 instead?
The proestrus rise in plasma LH is not registered. When were the animals sacrificed? If it was on the mornings of proestrus and estrus, LH plasma levels in Figure 6 are expected. However, the change in pituitary CNP content may not be registered if it happens in late proestrus. Does the pituitary Nppc/Npr2 expression change during the rat estrous cycle? It would have been nice to see male pituitary CNP content, in comparison to females.
Lines 409, 425; minor corrections of sentences needed.
Author Response
Reviewer 3
Line 72, reference 10. Should it be PMID: 7682940 instead?
Thank you for querying this. We chose the use the PMID: 10856898 reference, purely because it extends the initial description of the interaction between CNP and GnRH, with more pharmacological description of the desensitisation.
The proestrus rise in plasma LH is not registered. When were the animals sacrificed? If it was on the mornings of proestrus and estrus, LH plasma levels in Figure 6 are expected. However, the change in pituitary CNP content may not be registered if it happens in late proestrus.
This is a very important observation, and we apologise for not providing sufficient detail in the original manuscript. The animals were sacrificed on the morning of each stage of the cycle, so the proestrus surge in LH was not apparent. We have changed the description of the methods (p4, line 158) to clarify, and added sentences to the discussion (p13, lines467-470) to acknowledge the possible consequence of these timings.
Does the pituitary Nppc/Npr2 expression change during the rat estrous cycle? It would have been nice to see male pituitary CNP content, in comparison to females.
We have not performed those expression studies as yet, but agree that they will be very useful. We have previously published a comparison of pituitary CNP content in male and female Sprague Dawley and Wistar rats (Thompson et al., 2009) but failed to see any sexual dimorphic differences.
Lines 409, 425; minor corrections of sentences needed.
These have been modified, thank you for pointing these out.
Reviewer 4 Report
In this study, Mirczuk and colleagues investigate natriuretic peptide signaling in pituitary somatotropes. Their data indicate that Nppc expression in GH3 cells is responsive to both forskolin and TRH signaling through the PKA and MEK pathways respectively, and report putative novel downstream targets of CNP/Nppc signaling. Furthermore, the authors explore natriuretic peptide expression in feline somatotrope tumours and correlation between gene expression and clinical features. The experimental design appears rigorous and the manuscript is well-written.
Are the expression correlations in Figure 5 performed only on the HST samples? If so, I believe this data cannot be fully interpreted without the same analysis in controls, given that Figure 4 shows that NPPA, NPPC, NPR2, and NPR3 expression is not different between control and HST samples, suggesting that these correlations would also be observed in the controls. As a result, it is important to show whether the correlations between NPPC and D2R, GHR, and SSTR5 are specific to the tumour state or are correlated in the normal pituitary.
In my electronic version, Figure 1A is heavily pixellated and I am unable to read the axes labels, although I understand this may have occurred when was file was sent to me. Please verify that your original image is not pixellated. As a result, it is not clear to me what the peaks are in this figure and I can only infer that the x-axis is amplicon length in bp and the y-axis is intensity or similar. Line 185 indicates that the blue peaks are the named genes, but there are only 17 numbers and 29 listed genes. Supplemental Table S1 lists 15 non-cat primer sequences.
In Figure 1B, it is difficult to interpret whether the ~0.02 and ~0.04 expression of Nppa and Nppb relative to Actb are sufficient to indicate that these transcripts are being expressed in GH3 cells (line 173)? Can you include e.g. ct values or similar from the RT-qPCR for these genes to indicate that transcripts are readily detectible and only appear ‘low’ in comparison to the highly-expressed Actb? If this is what Fig 1A shows, please provide a list/table to indicate which genes are which numbers.
The experiments inhibiting of PKA and MEK signaling with forskolin and TRH treatment are sound but appear somewhat confirmatory because FSK and TRH are known to activate PKA and MEK pathways respectively (e.g. PMID 12080005), and Figure 2 may be more appropriate in the supplemental data.
Some genes in the heat maps in Figure 3A and 4A show very low expression relative to Actb and RPL18, making it difficult to interpret whether these genes are expressed in those cells/tissues, similar to my question about Fig 1B. While a 4-fold increase in cFos (Fig 3F) is convincing, the biological significance of a 0.5-fold reduction in Lepr (Fig 3J) is unclear if it is very lowly expressed in normal GH3 cells.
Line 372-373 states that your ‘data suggest enhanced expression of NPPA in feline pituitary tumour samples compared with normal…’, but Figure 4B and lines 266-7 indicate that NPPA, NPPB, and NPPC are not different between control and HST samples. Could you provide an explanation of the conclusion on line 372?
There are some spelling and grammar corrections e.g. lines 44-45 ‘expressed (in) feline pituitary tumours…expression was negative(ly) correlated’; line 425 ‘suggesting that cell-type specific effects of CNP [end of sentence]’.
Author Response
Reviewer 4
Are the expression correlations in Figure 5 performed only on the HST samples? If so, I believe this data cannot be fully interpreted without the same analysis in controls, given that Figure 4 shows that NPPA, NPPC, NPR2, and NPR3 expression is not different between control and HST samples, suggesting that these correlations would also be observed in the controls. As a result, it is important to show whether the correlations between NPPC and D2R, GHR, and SSTR5 are specific to the tumour state or are correlated in the normal pituitary.
Thank you for pointing this out, and many apologies for not being more clear in the original manuscript. The correlations reported are of gene expression levels in HST patient samples; we performed the same correlations in control samples, but failed to find any significance. We have added sentences in the results to clarify (p10, lines 335-557) and in the discussion (p12, lines 383-387), to expand upon the potential reasons for this.
In my electronic version, Figure 1A is heavily pixellated and I am unable to read the axes labels, although I understand this may have occurred when was file was sent to me. Please verify that your original image is not pixellated. As a result, it is not clear to me what the peaks are in this figure and I can only infer that the x-axis is amplicon length in bp and the y-axis is intensity or similar. Line 185 indicates that the blue peaks are the named genes, but there are only 17 numbers and 29 listed genes. Supplemental Table S1 lists 15 non-cat primer sequences.
We apologise for the poor resolution of this image. Unfortunately, we do not have access to our laboratory (since March 2020), where the GeXP machine and PC is located, which prevents us from replacing the image. Because we cannot replace this image at present, and as we have previously published representative electropherograms for these types of multiplex assays (Mirczuk et al. 2019, Scudder et al, 2019, both cited in the manuscript), we have decided to remove this image, and adjusted the figure (and legend) accordingly (p12, lines 187-195).
In Figure 1B, it is difficult to interpret whether the ~0.02 and ~0.04 expression of Nppa and Nppb relative to Actb are sufficient to indicate that these transcripts are being expressed in GH3 cells (line 173)? Can you include e.g. ct values or similar from the RT-qPCR for these genes to indicate that transcripts are readily detectible and only appear ‘low’ in comparison to the highly-expressed Actb? If this is what Fig 1A shows, please provide a list/table to indicate which genes are which numbers. Some genes in the heat maps in Figure 3A and 4A show very low expression relative to Actb and RPL18, making it difficult to interpret whether these genes are expressed in those cells/tissues, similar to my question about Fig 1B. While a 4-fold increase in cFos (Fig 3F) is convincing, the biological significance of a 0.5-fold reduction in Lepr (Fig 3J) is unclear if it is very lowly expressed in normal GH3 cells.
The multiplex RT-qPCR technology that we, and others, have employed takes advantage of fragment analysis to discriminate between transcripts of different size, measuring the fluorescent signal as a means of quantification. As such, this means multiple transcripts can be detected from a single sample, within the same PCR conditions. It is important to note that the chemistry does not allow for comparisons between different transcripts within the same sample – but it does allow for comparisons of the same transcripts between different samples. The main reason for this is that whilst the chemistry is adapted to permit the function of multiple different primer sets in the same reaction, under the same conditions, it is not possible to guarantee that each primer set functions with the same efficiency. Therefore, we do not make any statements about relative abundance of the different transcripts. All transcripts were expressed above the limit of detection, apart from Sstr4 which we failed to detect in rat pituitaries. In the case of the Lepr, the expression of this receptor in GH3 cells was first reported by Jin et al (2000; Endocrinology, Volume 141, Issue 1, 1 January 2000, Pages 333–339, https://doi.org/10.1210/endo.141.1.7260). Ultimately, these mRNA changes will eventually require investigation at the protein level.
The experiments inhibiting of PKA and MEK signaling with forskolin and TRH treatment are sound but appear somewhat confirmatory because FSK and TRH are known to activate PKA and MEK pathways respectively (e.g. PMID 12080005), and Figure 2 may be more appropriate in the supplemental data.
Thank you for raising this. To our knowledge, there has been no description that links regulation of endogenous Nppc mRNA expression to either MEK/ERK or cAMP/PKA signalling in pituitary cells (only our previous study in gonadotrope cell lines, that investigated signalling effects on the Nppc promoter (Thompson et al., 2009). Therefore, we think it is important to retain these data in the main body of the manuscript.
Line 372-373 states that your ‘data suggest enhanced expression of NPPA in feline pituitary tumour samples compared with normal…’, but Figure 4B and lines 266-7 indicate that NPPA, NPPB, and NPPC are not different between control and HST samples. Could you provide an explanation of the conclusion on line 372?.
We are very grateful to the reviewer for pointing out this typographical error; NPPA should have been NPR1, which is significantly elevated in HST patients. We have corrected this in the text at both places.
There are some spelling and grammar corrections e.g. lines 44-45 ‘expressed (in) feline pituitary tumours…expression was negative(ly) correlated’; line 425 ‘suggesting that cell-type specific effects of CNP [end of sentence]’. –
Thank you for pointing these out, we have corrected them.
Round 2
Reviewer 2 Report
I am happy with the answers to my queries.
Reviewer 4 Report
Many thanks for the author responses, which have appropriately addressed my comments.